# Inhibition of TNBC Cell Growth by Paroxetine: Induction of Apoptosis and Blockage of Autophagy Flux

**DOI:** 10.3390/cancers16050885

**Published:** 2024-02-22

**Authors:** Qianrui Huang, Mengling Wu, Yamin Pu, Junyou Zhou, Yiqian Zhang, Ru Li, Yong Xia, Yiwen Zhang, Yimei Ma

**Affiliations:** 1Department of Rehabilitation Medicine, State Key Laboratory of Biotherapy and Cancer Center, West China Hospital, Sichuan University and Collaborative Innovation Center for Biotherapy, Chengdu 610041, China; 2Department of Pediatrics, West China Second University Hospital, Sichuan University, Chengdu 610041, China; 3Key Laboratory of Birth Defects and Related Diseases of Women and Children, Ministry of Education, Sichuan University, Chengdu 610041, China; 4Department of Basic Medical Sciences & Forensic Medicine, West China Hospital, Sichuan University, Chengdu 610041, China; 5Innovation Center of Nursing Research, Nursing Key Laboratory of Sichuan Province, West China Hospital, Chengdu 610041, China; 6Key Laboratory of Rehabilitation Medicine in Sichuan Province/Rehabilitation Medicine Research Institute, Chengdu 610041, China

**Keywords:** triple-negative breast cancer (TNBC), paroxetine, apoptosis, autophagy, PI3K-Akt-mTOR pathway, combination therapy

## Abstract

**Simple Summary:**

The objective of drug repurposing is to discover novel therapeutic uses for established pharmaceuticals. Paroxetine (PX), a commonly prescribed antidepressant, has demonstrated promising anticancer properties in experimental studies. Nevertheless, its precise role and mechanisms of action in triple-negative breast cancer (TNBC) remain incompletely elucidated. Our research findings propose that PX may impact TNBC by modulating critical molecular pathways, providing significant implications for enhancing chemosensitivity and identifying potential therapeutic synergies in clinical practice. These findings provide a basis for the creation of individualized treatment plans and the identification of the most effective therapeutic interventions to enhance treatment precision and effectiveness in patients with TNBC.

**Abstract:**

The strategy of drug repurposing has gained traction in the field of cancer therapy as a means of discovering novel therapeutic uses for established pharmaceuticals. Paroxetine (PX), a selective serotonin reuptake inhibitor typically utilized in the treatment of depression, has demonstrated promise as an agent for combating cancer. Nevertheless, the specific functions and mechanisms by which PX operates in the context of triple-negative breast cancer (TNBC) remain ambiguous. This study aimed to examine the impact of PX on TNBC cells in vitro as both a standalone treatment and in conjunction with other pharmaceutical agents. Cell viability was measured using the 3-(4,5-dimethyl-2-thiazolyl)-2,5-diphenyl-2H-tetrazolium bromide (MTT) assay, apoptosis was assessed through flow cytometry, and the effects on signaling pathways were analyzed using RNA sequencing and Western blot techniques. Furthermore, a subcutaneous tumor model was utilized to assess the in vivo efficacy of combination therapy on tumor growth. The results of our study suggest that PX may activate the Ca^2+^-dependent mitochondria-mediated intrinsic apoptosis pathway in TNBC by potentially influencing the PI3K/AKT/mTOR pathway as well as by inducing cytoprotective autophagy. Additionally, the combination of PX and chemotherapeutic agents demonstrated moderate inhibitory effects on 4T1 tumor growth in an in vivo model. These findings indicate that PX may exert its effects on TNBC through modulation of critical molecular pathways, offering important implications for improving chemosensitivity and identifying potential therapeutic combinations for clinical use.

## 1. Introduction

Breast cancer is a prevalent and concerning disease affecting women today, with an estimated 1.6 million cases annually and a rising incidence [1,2]. It ranks third in mortality after lung cancer and colorectal cancer, underscoring its substantial impact on women’s health [2]. Triple-negative breast cancer (TNBC) is a molecular subtype of breast cancer characterized by the absence of estrogen receptors, progesterone receptors, and human epidermal growth factor receptor 2 expression [3,4]. TNBC accounts for 15–20% of breast cancer cases and is distinguished by its high invasiveness and malignancy, presenting a challenging prognosis compared to other subtypes [5]. Current treatment modalities for breast cancer encompass both localized and systemic approaches, such as surgery and chemotherapy [5]. Nevertheless, the effectiveness of chemotherapy is constrained, and the development of drug resistance often leads to TNBC recurrence and worse patient outcomes [6]. Overcoming these obstacles necessitates the exploration of novel therapeutic approaches, potentially through synergistic combinations of established medications.

Paclitaxel and anthracycline-based chemotherapy regimens have long been established as standard treatment options for patients with TNBC [2,7,8]. Among these regimens, doxorubicin (DOX), an anthracycline agent, is known to inhibit nucleic acid synthesis through DNA intercalation, thereby impeding cancer cell proliferation [9]. Nevertheless, the efficacy of DOX as a standalone treatment is limited by the development of adaptive cellular responses over time, leading to the acquisition of drug resistance. TNBC frequently exhibits resistance to DOX, leading to the emergence of various multidrug-resistant phenotypes in cancer cells [10,11]. Moreover, the application of DOX is hindered by multicellular toxicities, particularly cardiac toxicity, which limits its anticancer potential [9]. Consequently, enhancing the efficacy of DOX is crucial to developing safer treatment approaches for TNBC.

Recently, the complex relationship between cancer and emotional states has been found to be more nuanced. Notably, depression has been found to exhibit a significant association with various types of cancer, including oral and pharyngeal cancers (22–57%), pancreatic cancer (33–50%), breast cancer (1.5–46%), and lung cancer (11–44%) [12]. Particularly noteworthy is the heightened vulnerability of individuals with breast cancer to depression, attributed to the profound psychological impact of changes in body image, which can ultimately compromise both quality of life and prognosis [13]. A survey conducted by the American Cancer Society emphasizes the impact of depression on adherence to cancer treatment, underscoring the importance of psychological interventions for cancer patients experiencing depression [14]. The integration of antidepressants as a treatment modality plays a significant role in addressing depression within the cancer population. A number of studies have suggested potential synergy between antidepressants and tamoxifen, possibly leading to a reduction in breast cancer recurrence [15,16], although Haque et al. presented contrasting perspectives on this matter [17]. Considering the intricate interplay between antidepressants and anticancer drugs, the fusion of antidepressant interventions with inherent anticancer attributes presents a promising avenue to mitigate the cascade of adverse effects often associated with cancer treatment in individuals with depression [17].

In recent years there has been an increasing acknowledgment of the practice of drug repurposing, which offers notable advantages in terms of time and cost efficiency compared to traditional drug discovery methods. Numerous studies have highlighted the potential anticancer properties of antidepressants, including paroxetine, in various types of tumors. Paroxetine (PX) is an effective selective serotonin reuptake inhibitor, and current research suggests its potential anticancer properties in breast and colorectal tumors [18,19]. Additionally, PX has been reported to inhibit DNA synthesis in malignant lymphomas [20] and to exhibit cytotoxic effects on a variety of malignant tumors [21,22]. Earlier investigations have shown that PX induces pro-apoptotic activity in liver cancer cells and gastric cancer cells [23,24]. Recent research suggests that PX may modulate the potential cancer therapy target GRK and exhibit immunotherapeutic effects in malignant tumors [25,26]. It is noteworthy that its anticancer activity in TNBC remains unclear. Therefore, our research concentrates on the synergistic use of PX and DOX in TNBC therapy, with a focus on elucidating the underlying mechanistic rationale.

## 2. Materials and Methods

### 2.1. Chemicals and Reagents

Paroxetine hydrochloride was purchased from J&K Scientific (Beijing, China); doxorubicin hydrochloride, chloroquine (CQ), and 3-Methyladenine (3-MA) were purchased from Aladdin (Shanghai, China); 3-(4,5-dimethythiazol)-2,5-diphenyltetrazolium bromide (MTT) was purchased from CSNpharm (Chicago, IL, USA); the Annexin V-PE/7-AAD Apoptosis Detection Kit was purchased from BioLegend (San Diego, CA, USA); DCFH-DA was purchased from MCE (Monmouth Junction, NJ, USA); rhodamine 123 was purchased from Sigma-Aldrich (St. Louis, MO, USA); fluo 3-AM was purchased from Beyotime (Shanghai, China); lipo2000 was purchased from Invitrogen (Carlsbad, CA, USA); the BCA Protein Assay Kit was purchased from Beyotime (Shanghai, China); cleaved-caspase3 and PARP antibodies were purchased from Cell Signaling Technology (Boston, MA, USA); P62, LC3, BAX, BAD, Bcl-2, Bcl-xl, and Mcl-1 antibodies were purchased from Abways Technology (Shanghai, China); cIAP-1, cIAP-2, AKT, and p-AKT antibodies were purchased from ABclonal Technology (Wuhan, China); Beclin1, mTOR, and p-mTOR antibodies were purchased from ABmart Inc. (Shanghai, China); chemiluminescence reagents were purchased from Amersham (Piscataway, NJ, USA); tribromoethanol was purchased from Sigma-Aldrich (St. Louis, MO, USA); and D-luciferin potassium was purchased from Abcam (Cambridge, MA, USA).

Paroxetine was dissolved in DMSO to prepare a stock solution and then diluted with DMEM to achieve the final working concentration for all experiments. To avoid cell toxicity, the final concentration of DMSO in all treatment groups was <0.1%.

### 2.2. Cell Lines and Cell Cultures

The TNBC cell lines 4T1 (murine) and MDA-MB-231 (human) were procured from the American Type Culture Collection (Manassas, VA, USA) and preserved in frozen storage until study commencement. The construction of 4T1-luc was executed within our laboratory. MCF-10A was procured from MeisenCTCC (Hangzhou, China). TNBC cells were sustained in DMEM medium supplemented with 10% FBS and 1% penicillin-streptomycin-glutamine. All cell cultures were maintained at 37 °C in a humidified atmosphere enriched with 5% CO_2_.

### 2.3. Cell Viability Assay

Cell viability was assessed utilizing MTT assays. TNBC cells were evenly distributed within a 96-well plate at a logarithmic growth density of 1–5 × 10^3^ cells and incubated overnight. Subsequent to this, cells were exposed to diverse concentrations of agents for 24 h, 48 h, and 72 h durations. Following treatment, each well received 20 μL of MTT reagent (5 mg/mL), with subsequent 2–4 h incubation before removal of medium. This was followed by the addition of 150 μL dimethyl sulfoxide (DMSO) per well with thorough mixing. Cell viability was quantified by measuring absorbance at 570 nm using a Spectra Max M5 microplate reader (Molecular Devices, San Jose, CA, USA). Each assay was executed in triplicate.

### 2.4. Colony Formation Assay

First, 4T1 or MDA-MB-231 cells were seeded in 6-well plates at a density of 1000 cells per well and cultivated for 24 h. Cells were then exposed to PX, DOX alone, or their combination. Following 7–14 days of incubation, colonies were fixed with 4% paraformaldehyde and stained with 0.5% crystal violet, then colonies consisting of >50 cells were captured and enumerated utilizing ImageJ software (version 1.8.0). Each assay was performed in triplicate.

### 2.5. Apoptosis Assays by Flow Cytometry

Apoptosis was evaluated using the Annexin V-PE/7-AAD Apoptosis Detection Kit (BioLegend, San Diego, CA, USA) as previously detailed [27]. Following seeding of 4T1 and MDA-MB-231 cells in 6-well plates and overnight incubation, cells were subjected to PX, ZVAD-FMK, CQ, 3-MA, DOX alone, or their combination for stipulated time frames. Flow cytometric analysis was then employed to quantify the percentage of apoptotic cells. Each assay was conducted in triplicate.

### 2.6. Analysis of Reactive Oxygen Species (ROS)

Intracellular ROS levels were assessed via DCFH-DA (MCE, Monmouth Junction, NJ, USA) assay as previously outlined [28]. Briefly, adherent cells treated with varying agent concentrations and durations were stained with DCFH-DA (10 μM) for 30 min. Relative ROS levels were quantified using flow cytometry. Each assay was performed in triplicate.

### 2.7. Analysis of Mitochondrial Membrane Potential (Δψm)

Measurement of Δψm was accomplished through Rhodamine123 (Rh123) (Sigma-Aldrich, St. Louis, MO, USA) and flow cytometry as previously described [29]. After treatment with indicated agents, adherent cells were stained with Rh123 (5 μg/mL) for 15–30 min, followed by flow cytometry assessment of ΔΨm loss. Each assay was conducted in triplicate.

### 2.8. Analysis of Calcium Mobilization

Calcium mobilization was gauged using Fluo 3-AM (Beyotime, Shanghai, China), a fluorogenic probe [30]; adherent cells were treated with indicated agents, stained with Fluo 3-AM (0.5 μM) for 30 min, and subjected to flow cytometry analysis of calcium mobilization. Each assay was executed in triplicate.

### 2.9. Drug Synergy Studies and Data Analysis

The Chou–Talalay methodology was employed to assess drug synergy [30]. Cells were seeded in 96-well plates at densities ranging from 1 to 5000 cells/well and treated with PX and DOX alone or in combination for 24 h, 48 h, and 72 h. CompuSyn software (version 1.0) was utilized to compute combination index (CI) values based on the Chou–Talalay method, with CI values greater than 1 indicating antagonism, CI values less than 1 denoting synergy, and a CI value equal to 1 indicating an additive effect [31].

### 2.10. Western Blot Analysis

Following treatment of TNBC cells with the specified reagents, alterations in protein expression profiles were assessed using Western blot analysis as previously outlined [32]. Subsequent to cell harvesting, cellular lysis was executed employing a cell lysate augmented with a protease inhibitor cocktail and maintained under refrigeration. The resulting lysate was subjected to centrifugation at 13,000 rpm for a duration of 15 min employing a low-temperature centrifuge. Protein content was determined utilizing a BCA Protein Assay Kit (Beyotime, Shanghai, China) while adhering to established protocols. Samples were then separated through 12% SDS-PAGE and transferred to polyvinylidene difluoride (PVDF) membranes (Merck Millipore, Billerica, MA, USA). Before commencement of primary antibody incubation, the membranes were precluded from nonspecific binding by blocking with a 5% skim milk solution in TBS/T for a span of 1 h. Incubation with primary antibodies was conducted overnight at 4 °C, succeeded by successive washing steps with TBS/T and incubation with secondary IgG antibodies labeled with horseradish peroxidase (HRP) carried out at room temperature for 1 h. Visualization of protein bands was achieved through the utilization of chemiluminescence reagents.

### 2.11. RNA Sequencing (RNA-Seq)

RNA-Seq analysis was performed by OE Biotech, Inc. (Shanghai, China). Briefly, 4T1 cells were subjected to treatment with either a control or 20 μM PX for 24 h. Three samples were collected for each group, and total RNA was extracted using Trizol reagent. The purity and quantity of RNA were assessed using a NanoDrop 2000 spectrophotometer (Thermo Scientific, Waltham, MA, USA). Transcriptome libraries were prepared in accordance with the provided kit instructions and subsequently sequenced on the Illumina Novaseq 6000 platform (San Diego, CA, USA) following the manufacturer’s guidelines. DESeq2 (version 1.30.0) software was employed for differential genes expression analysis, with genes meeting the criteria of q-value < 0.05 and fold change >2 or fold change <0.5 being classified as differentially expressed genes (DEGs).

### 2.12. Transient Transfection and Immunofluorescence Analysis

Adherent cells underwent transfection with LC3-GFP-RFP utilizing Lipo 2000 (Invitrogen, Carlsbad, CA, USA) as a transfection agent. Following an incubation period of 6 h, the medium was replaced; 24 h post-transfection, cells were harvested and seeded onto Millicell EZ SLIDES (Merckmillipore, Billerica, MA, USA) before being subjected to treatment with PX and CQ for a duration of 24 h. Subsequently, cells were fixed using 4% paraformaldehyde for 30 min at room temperature and rinsed with PBS. Nuclear staining was performed using DAPI solution (5 μg/mL) for 5 min at 37 °C. Visualization of LC3-GFP-RFP transfected cells was achieved through confocal laser microscopy.

### 2.13. In Vivo Murine Experiments

The in vivo murine investigations were executed within the State Key Laboratory of Biotherapy and Tumor Center at Sichuan University following ethical approval from the institution’s Ethics Committee. Female BALB/c mice with an age range of 8–10 weeks were procured from HFK Bioscience (Beijing, China) to serve as the experimental cohort. Subcutaneous 4T1 tumors were initiated by inoculating 1 × 10^5^ luciferase-expressing 4T1 cells into the left flank of each murine subject. Upon achieving an approximate mean tumor volume of 100 mm^3^ (±S.E.M), the mice were subjected to random assignment into six distinct treatment categories: (1) Vehicle, (2) PX, (3) DOX, (4) PX + DOX, (5) CPB + DOC, and (6) CPB + DOC + PX. The administration of PX occurred at a prescribed dosage of 20 mg/kg once daily via intraperitoneal injection (i.p.), while DOX was introduced at a dosage of 5 mg/kg administered once every 6 days through intravenous injection (i.v.). The additional drug combination dosing regimen involved CBP administered intraperitoneally every 6 days at a dose of 8 mg/kg and DOC administered intravenously every 6 days at a dose of 10 mg/kg. Intervention, encompassing tumor resection and metastasis evaluation, was performed when tumor volumes reached 650 mm^3^. For tumor resection, all groups of mice were anesthetized using tribromoethanol (60 mg/kg i.p.). A meticulously structured dosing regimen, illustrated in Section 3.7 guided the therapeutic interventions. To ensure precise allocation, cage cards detailing treatment group designations were securely affixed to each cage following the process of randomization. It is noteworthy that mice subjected to drug administration were properly segregated from their control-treated counterparts. Throughout the treatment phase, drug dispensation followed a systematic two-cage-at-a-time protocol, starting with the vehicle-treated mice. Following the completion of drug administration to the vehicle-treated mice, the treated counterparts were subsequently relocated from their respective cages. The computation of tumor volume was executed employing the formula $V\;\left({\mathrm{mm}}^{3}\right)=L\;\left(\mathrm{mm}\right)\times W^{2}\;\left({\mathrm{mm}}^{2}\right)\times 0.5$, accompanied by consistent recording of tumor volume measurements and body weights at intervals spanning 2–3 days.

### 2.14. Statistical Analysis

The acquired data were exhibited in the format of mean ± standard deviation (S.D.) or mean ± standard error of the mean (S.E.M.) originating from distinct independent replicates, and were subsequently subjected to comprehensive analysis employing GraphPad Prism 9.0 software. Comparative assessments between two distinct groups were conducted utilizing either the Student’s *t*-test or the Mann–Whitney U-test as contextually appropriate. Statistically significant findings are appropriately indicated utilizing the following notation: *** *p* < 0.001, ** *p* < 0.01, and * *p* < 0.05, or alternatively, ### *p* < 0.001, ## *p* < 0.01, and # *p* < 0.05.

## 3. Results

### 3.1. Inhibition of TNBC Cell Line Growth In Vitro by PX

Recent investigations have demonstrated the antineoplastic properties of paroxetine (PX) on various types of cancer cells, encompassing lung [33], colorectal [19], mammary gland [18], and gastric [24] malignancies. However, the potential relationship between PX and TNBC has not been investigated. Therefore, our research aims to investigate the effects of PX on TNBC. The chemical structure of PX is shown in Figure 1A. In the initial phase, we evaluated the immediate impact of transient treatment (24–72 h) on the proliferation of two TNBC cell lines, 4T1 and MDA-MB-231, employing an MTT assay. The pharmacodynamic influence of PX on TNBC cells was analyzed at various time points and concentrations ranging from 3.75–30 μM, as graphically depicted in Figure 1B. The values denoting the half-maximal inhibitory concentration (IC50) of PX for the 4T1 and MDA-MB-231 cell lines at 24 h, 48 h, and 72 h were determined to be 19.44 μM, 13.34 μM, and 7.63 μM for 4T1 and 22.3 μM, 19.38 μM, and 7.88 μM for MDA-MB-231 (Figure 1C). The inhibitory effects of PX on the viability of 4T1 and MDA-MB-231 cells exhibited a concentration-dependent trend. In contrast, PX demonstrated reduced cytotoxicity towards human normal breast epithelial cells MCF-10A (Figure 1D). This observation implies that PX may possess enhanced cytotoxic effects on TNBC cell lines relative to non-neoplastic breast cells, indicating a degree of selectivity in its ability to diminish the viability of TNBC cells. To further validate the inhibitory influence of PX treatment on TNBC cell lines, a colony formation assay was conducted, revealing that PX can hinder cellular proliferation at low concentrations (Figure 1E,F). In conclusion, these outcomes postulate a plausible cytotoxic effect of PX on TNBC cell lines.

### 3.2. Induction of Apoptosis in TNBC Cell Lines by PX In Vitro

Previous studies have highlighted the ability of PX to inhibit cancer cell growth by inducing apoptosis. We hypothesized that PX treatment could induce apoptosis in TNBC cell lines. Following treatment with various concentrations of PX for 24 h, apoptotic events in 4T1 and MDA-MB-231 cells were analyzed using flow cytometry. The percentage of apoptotic cells in both cell populations showed a dose-dependent increase after PX treatment (Figure 2A,C).

The initiation of caspases and the subsequent cascade activation of downstream caspases are essential components of the cellular apoptosis process [34]. The pan-caspase inhibitor Z-VAD-FMK was employed in tandem with PX administration to 4T1 and MDA-MB-231 cells, revealing a partial reduction in PX-induced apoptosis in 4T1 cells (Figure 2B,D). These results collectively demonstrate that PX has the potential to partially reduce TNBC cell viability through caspase-dependent apoptotic pathways.

### 3.3. Induction of Apoptosis in TNBC Cell Lines by PX via Activation of the Mitochondria-Mediated Intrinsic Pathway

Apoptosis, a regulated form of cellular demise, can be triggered via either the extrinsic pathway, involving death receptors, or the intrinsic mitochondrial pathway [35,36]. Among vertebrate cells, the intrinsic mitochondrial pathway is the predominant mechanism of apoptosis. Activation of this pathway is prompted by internal apoptotic stimuli such as oncogenic activation, DNA damage, hypoxia, and growth factor deprivation [36]. The regulation of the intrinsic apoptosis pathway involves the modulation of mitochondrial outer membrane permeability by Bcl-2 family proteins, which in turn modulates ΔΨm [37]. Additionally, substantial accumulation of intramitochondrial Ca^2+^ plays a role in intrinsic apoptosis [38,39,40]. Our experimental study included the evaluation of intracellular levels of ROS, ΔΨm, and Ca^2+^ concentrations. Relative to the control group, cells subjected to PX treatment demonstrated a concentration-dependent increase in relative ROS levels and mitochondrial Ca^2+^ concentration (Figure 2E,G,H,J). Additionally, the decline in ΔΨm in TNBC cells under PX treatment showed a heightened significance with increasing concentrations (Figure 2F,I).

In pursuit of unraveling the apoptotic induction mechanism by PX in TNBC cells, we scrutinized the expression levels of proteins integral to intrinsic apoptosis. Caspases, a class of cysteine proteases, play a central role in the execution of cellular apoptosis. Inhibitor of apoptosis proteins (IAPs) act to inhibit caspase activity, and overcoming this inhibition indirectly facilitates apoptosis [41]. The B-cell lymphoma 2 (Bcl-2) family-regulated pathway, which governs mitochondrial outer membrane permeabilization (MOMP), is a key component of the intrinsic apoptosis pathway. In this particular family, proteins can be categorized as either pro-apoptotic or anti-apoptotic based on their structural and functional attributes, with the presence of shared BH3 structural domains facilitating complex formation [42,43,44]. The initiation of intrinsic apoptosis can be attributed to the activation or post-translational modification of BH3-only proteins, leading to the inhibition of specific BCL-2 family anti-apoptotic proteins (Bcl-2, Bcl-xl, Mcl-1). Consequently, this release of inhibition allows for the activation of the pro-apoptotic proteins BAX and BAK, ultimately promoting apoptosis [44]. In light of the concentration-dependent effects of PX on TNBC, we investigated variations in the levels of intrinsic pathway-associated proteins following distinct PX concentrations. Upon treating 4T1 and MDA-MB-231 cells with varied PX concentrations for 24 h, noticeable changes in the expression of apoptosis-related proteins were observed. Our findings revealed that PX treatment led to an increase in the levels of cleaved caspase 3 and cleaved PARP, both of which are well-known markers of apoptosis (Figure 2K). In addition, we found that Caspase 9, a key player in the initiation of the intrinsic apoptotic pathway [45], was increased following PX treatment, suggesting the activation of the intrinsic apoptotic pathway in TNBC cells (Figure 2K). Examination of IAP protein expression unveiled a decrease in cIAP-1 and cIAP-2 levels with increasing concentrations of PX (Figure 2K). Evaluation of Bcl-2 family protein expression indicated that levels of the anti-apoptotic proteins BAX and BAD increased in 4T1 cells as the PX concentration increased. In contrast, the downregulation of pro-apoptotic factors such as Bcl-2, Mcl-1, and Bcl-xl resulted in an increase in the BAX/Bcl-2 and BAD/Bcl-xl ratios (Figure 2K). Analogously, in MDA-MB-231 cells, elevated BAX/Bcl-2 and BAD/Bcl-xl ratios and decreased expression of Mcl-1 mirrored the protein expression patterns observed in 4T1 cells (Figure 2K). These findings collectively suggest that PX may potentially inhibit TNBC cell vitality through instigation of the mitochondria-mediated intrinsic pathway.

### 3.4. Induction of Cytoprotective Autophagy and Autophagic Flux Inhibition by PX in TNBC Cell Lines

Following this, we conducted an investigation to elucidate the potential mechanisms responsible for the inhibition of TNBC by PX. After treatment with PX, 4T1 cell lines were collected for RNA-Seq analysis. DEGs were ascertained employing rigorous criteria (Padj < 0.05, log2FC [fold change] > ±1), unveiling a total of 2899 DEGs. Among these, 1478 genes were upregulated, while 1421 were downregulated (Figure 3A). A comprehensive investigation utilizing KEGG pathway enrichment analysis was conducted with the assistance of the DESeq2 bioinformatics tool, revealing an array of downstream signaling pathways that may be influenced by PX intervention (Figure 3B). Importantly, autophagy emerged as a prominently enriched pathway following PX treatment, as illustrated within the red box in Figure 3B. Furthermore, gene set enrichment analysis (GSEA) underscored a significant upregulation of mitochondrial autophagy-associated pathways in the PX treatment group compared to the Vehicle group (Figure 3C). This significant observation served as the impetus for our further investigation into the field of autophagy.

Autophagy plays a dual role during tumor development, with an appropriate level of autophagic activity supporting cellular viability in oncological interventions and excessive activity potentially leading to cellular apoptosis [46,47]. In order to validate the observations derived from RNA-seq, an investigation was undertaken to explore the dose-dependent influence of PX on autophagy in TNBC cell lines. The conversion of LC3-I to LC3-II, a well-established indicator of autophagosome formation, was demonstrated to occur in a PX concentration-dependent manner (Figure 3D). Additionally, chloroquine (CQ) was employed as a positive control to induce autophagosome accumulation, and subsequent immunofluorescence analyses confirmed the PX-induced increase in LC3 puncta formation (Figure 3E). The status of autophagic flux can be categorized as either normal or impeded, with discernible alterations reflected through modulations in the expression levels of the autophagy-specific substrate P62 [48]. An elevation in P62 protein levels was evident in both 4T1 and MDA-MB-231 cells following treatment with PX (Figure 3D), suggesting that PX can potentially impede autophagic flux. To investigate whether this compromised autophagic flux could be attributed to a disruption in autophagosome–lysosome fusion, 4T1 cells were transfected with tandem RFP-GFP-tagged LC3B plasmids. Using CQ as a positive control for autolysosome inhibition, it was observed that the majority of LC3B fluorescent puncta exhibited a yellow hue (RFP+GFP+ signal, signifying autophagosomes) following treatment with PX, as opposed to a red puncta (RFP+GFP− signal, indicating autolysosomes) (Figure 3F). This phenomenon indicates that PX treatment led to an increase in autophagosome accumulation while simultaneously reducing the formation of autolysosomes.

The experimental study mentioned above provides compelling evidence that PX significantly inhibits autophagic flux in TNBC cells. To further understand the role of autophagy in the anti-tumor effects of PX in the 4T1 and MDA-MB-231 cell lines, we employed a sequential approach involving co-treatment with PX and two different autophagy inhibitors: CQ, an inhibitor of autophagosome–lysosome fusion, and 3-MA, recognized for its capability to impede the initiation of autophagy via inhibition of PI3K, were employed concomitantly. Our comprehensive dataset unequivocally indicates that the suppressive impact of PX on TNBC cell lines is significantly enhanced through the combined use of autophagy inhibitors (Figure 3G). Furthermore, thorough analyses conducted through flow cytometry definitively illustrate that concurrent administration of either CQ and PX or 3-MA and PX leads to a notable augmentation in apoptotic occurrences in TNBC cell lines (Figure 3H–K).

In summary, the findings of our study offer compelling evidence supporting the capacity of PX to induce autophagy in TNBC cell lines, thereby influencing its involvement in the progression of TNBC. This mechanism is believed to be accomplished by inhibiting autophagic flux, underscoring the significance of modulating autophagy in order to augment the therapeutic effectiveness of PX in combating TNBC.

### 3.5. Inhibition of TNBC by PX via the PI3K-AKT-mTOR Pathway

Moreover, a KEGG analysis of DEGs meeting criteria of adjusted *p*-values below 0.05 and log2 fold change equal to or exceeding ±1 revealed significant enrichment of the PI3K-AKT and mTOR pathway as a key signaling cascade linked to PX (Figure 4A). Dysregulated activation of the PI3K-AKT-mTOR pathway is frequently reported in cancer research, and mTOR is implicated in chemotherapy resistance as well, prompting our exploration of this pathway [49,50,51]. The expression of relevant proteins in TNBC cell lines was observed following treatment with PX. Notably, the experimental data presented in Figure 4B indicate that PX treatment led to a decrease in the expression of phosphorylated AKT and phosphorylated mTOR proteins in both 4T1 and MDA-MB-231 cells, suggesting an inhibitory effect of PX on the PI3K-AKT-mTOR pathway. Consequently, it is postulated that PX may exert its inhibitory actions on TNBC via modulation of the PI3K-AKT-mTOR signaling pathway.

### 3.6. Potentiation of TNBC Cancer Cell Sensitivity to DOX Treatment by PX In Vitro

DOX is a frontline chemotherapeutic agent used in the management of TNBC, and demonstrates significant antitumor efficacy. Regrettably, the clinical use of DOX is hindered by notable cytotoxic effects, particularly cardiotoxicity, limiting its therapeutic utility [9]. This challenge underscores the need for the development of effective combination strategies that preserve the anticancer properties of DOX while mitigating its toxicity. In this context, our study explored the potential of PX to augment the antitumor effects of DOX. The cytotoxic effects of the PX-DOX combination on TNBC cell lines were assessed through MTT assay, revealing distinct drug interactions at varying concentrations. Our findings revealed that PX and DOX synergistically suppressed the viability of both 4T1 and MDA-MB-231 cells to a certain degree (Figure 5A). We quantified the synergistic impact utilizing CompuSyn software (version 1.0), with the combination consistently yielding CI values below 1 for various concentration pairs (Figure 5B), suggesting a potential synergistic relationship between the two compounds. To optimize the combined therapeutic impact, we selected a drug concentration combination with a low concentration and CI value below 1. Verification through colony formation assays further corroborated the curtailed viability of 4T1 cells when exposed to both DOX and PX (Figure 5C). Collectively, these outcomes underscore the potential of PX to enhance the antitumor efficacy of DOX.

Both DOX and PX have been shown to induce antineoplastic effects by promoting apoptosis in malignant cells [52]. This study investigated the potential of PX to enhance DOX-induced apoptosis. The 4T1 and MDA-MB-231 cell lines were subjected to DOX, PX, or their combination for 48 h. The apoptosis rates for 4T1 cells when treated with DOX (0.125 μM) or PX (12.5 μM) alone were 14.48% and 21.75%, respectively. Correspondingly, the rates for MDA-MB-231 cells subjected to DOX (0.125 μM) or PX (12.5 μM) alone were 29.46% and 23.53%, respectively. Strikingly, the combined treatment propelled the apoptosis rates in 4T1 and MDA-MB-231 cells to 41.16% and 60.63%, respectively, nearly doubling or tripling those observed in the single-drug groups (Figure 5D,E). Furthermore, DOX is known to induce cellular apoptosis via disruption of mitochondrial respiration, which is associated with protein synthesis and lipid peroxidation and is facilitated by the accumulation of intracellular ROS, cytochrome c release, and caspase family activation. Our findings demonstrate that the combined treatment elevated intracellular ROS levels approximately twofold compared to DOX or PX monotherapy (Figure 5F,G). Additionally, the intracellular Ca^2+^ levels in 4T1 cells were increased twofold in the combination treatment group compared to the single-drug regimens (Figure 5H). In conclusion, our research highlights the ability of PX to enhance the inhibition of TNBC induced by DOX to some extent.

### 3.7. Potentiation of TNBC Tumor Sensitivity to Chemotherapy In Vivo by PX

In vitro investigations have established the capability of PX to augment the sensitivity of TNBC cells to DOX treatment. Building upon these results, we conducted further research to investigate the synergistic interplay between DOX and PX within an in vivo animal model. Previous studies have demonstrated that PX can inhibit the PI3K-Akt-mTOR pathway in vitro. Upregulation of the AKT-mTOR signaling pathway in platinum-resistant tumor cells has been documented and targeting mTORC1/2 has been shown to reverse platinum resistance in tumors, guiding the direction of our investigation [53]. In accordance with findings from the previous KEGG analyses, notable alterations in platinum resistance were observed in TNBC following PX treatment (Figure 4A). Importantly, the concurrent administration of carboplatin (CBP) and docetaxel (DOC) represents a prevalent clinical protocol for TNBC. Therefore, while establishing the 4T1 subcutaneous tumor model for PX combined with DOX treatment, two supplementary groups were incorporated to evaluate the efficacy of CBP + DOC and PX combination therapy in TNBC (Figure 6A). The experimental procedure, depicted in Figure 6A, involved the inoculation of 1 × 10^5^ 4T1-luc cells into the left flank of mice to induce subcutaneous 4T1 tumors. Upon reaching an average tumor volume of approximately 100 mm^3^ (±S.E.M), the mice were randomly assigned to one of six distinct treatment groups: (1) Vehicle, (2) PX, (3) DOX, (4) PX + DOX, (5) CPB + DOC, and (6) CPB + DOC + PX. PX was administered intraperitoneally (i.p.) daily at a dose of 20 mg/kg, while DOX was administered intravenously every six days at a dose of 5 mg/kg. The supplementary drug combination dosing regimen included intraperitoneal administration of CBP every 6 days at a dosage of 8 mg/kg and intravenous administration of DOC every 6 days at a dosage of 10 mg/kg.

Our study findings demonstrated that PX and DOX exhibit some synergistic inhibitory effects on tumor growth in vivo. Concurrent administration of PX and DOX demonstrated a modest inhibitory impact on tumor growth (Figure 6B) while maintaining body weight levels (Figure 6C). The combination of PX with CBP and DOC exhibited a slight inhibitory effect compared to singular treatments (Figure 6D), with no notable changes in body weight (Figure 6E). Compared to individual drug treatments, the combined drug regimen did not yield statistically significant differences in inhibition (Figure 6D). These data suggest that PX might enhance the anti-tumor efficacy of TNBC chemotherapeutic agents.

## 4. Discussion

This study provides valuable insights into the potential of PX to inhibit the growth and viability of TNBC cells, underscoring its therapeutic potential for TNBC treatment when used in conjunction with DOX. Our results demonstrate the ability of PX to induce apoptosis and enhance autophagy in TNBC cells by targeting the PI3K/AKT/mTOR signaling pathway, underscoring its potential as a therapeutic agent for TNBC.

Despite continued research and development, clinical outcomes for TNBC treatment have not reached optimal levels. DOX continues to be a vital therapeutic choice for TNBC; however, its extended use is limited by concerns around resistance and cardiotoxicity. The strategy of drug repurposing has gained considerable attention in recent times, with various studies demonstrating a correlation between antidepressants and cancer [54,55,56]. Additionally, several clinical trials, such as NCT06225011 and NCT02217709, have explored the potential of using antidepressants for cancer treatment. Therefore, the utilization of antidepressants for alternative therapeutic purposes carries substantial clinical significance. Paroxetine (PX), a commonly prescribed antidepressant, has been found to exhibit potential anticancer properties In multiple types of malignancies. Serafeim et al. documented the ability of PX to impede DNA synthesis in Burkitt’s lymphoma biopsy samples [20]. Levkovitz et al. revealed its growth-inhibitory effects on neuroblastoma cell lines, which were accompanied by induction of apoptosis [21]. Kuwahara et al. demonstrated that PX can augment caspase 3/7 activity in HepG-2 hepatocellular carcinoma cells, leading to enhanced apoptosis and improved antitumor effectiveness [23]. Jang et al. illuminated PX-induced apoptosis and diminished cell viability in the HCT116 and HT-29 colorectal cancer cell lines [19]. Moreover, Liu et al. presented results on the suppression of autophagy by PX, resulting in heightened ROS accumulation and aggravated DNA damage in AGS gastric cancer cell lines [24]. Consistent with these findings, our study shows that PX is effective in stopping cell growth, promoting autophagy, and causing cell death and that it displays anti-cancer properties in TNBC, while additionally uncovering its potential mechanism.

A pivotal facet warrants consideration here, namely, the role of apoptosis in the realm of cancer therapy. Apoptosis, a programmed cell death mechanism, plays a crucial role in preserving tissue homeostasis and eliminating aberrant or impaired cells [57]. Dysregulation of apoptosis is a cardinal hallmark of cancer, including TNBC, prompting the exploration of apoptosis induction as a therapeutic modality. The activation of caspase3 and PARP cleavage are pivotal events in the apoptotic pathway. This process can be instigated via two principal routes: the extrinsic pathway, which involves the interaction of cell surface death receptors (e.g., Fas) with their corresponding ligands, and the intrinsic pathway, which is controlled by Bcl-2 family proteins that modulate mitochondrial permeability to regulate apoptosis [58]. Various stimuli can activate the intrinsic apoptotic pathway, leading to changes in the mitochondrial inner membrane and the opening of the mitochondrial permeability transition pore, ultimately resulting in the loss of mitochondrial membrane potential, which in turn leads to the release of cytotoxic mitochondrial components upon cellular death. It is noteworthy that Bcl-2 family proteins are found in the endoplasmic reticulum (ER) as well; the release of ER-derived calcium assumes a pivotal status in numerous processes related to apoptosis. External stimuli can disrupt the balance of Ca^2+^ within the ER, leading to an increase in cytoplasmic and mitochondrial Ca^2+^ levels, which may impact the activities of mitochondrial and Bcl-2 family proteins and ultimately direct cells towards apoptosis [59,60]. Our recent research results confirm that PX treatment increases intracellular ROS levels, leading to a significant decrease in mitochondrial membrane potential and disruption of the Ca^2+^ balance in TNBC cells, suggesting compromised integrity of the mitochondrial membrane. The altered expression patterns of caspases and IAPs provides further evidence for this conclusion. Therefore, our findings suggest that PX has the potential to induce intrinsic apoptosis through a caspase-dependent and mitochondria-mediated pathway.

Additionally, this study underscores the significance of autophagy in malignancy. KEGG analysis of RNA-seq data subsequent to PX treatment implies potential involvement of PX in the autophagy regulatory network. Autophagy, a cellular process involved in degradation and recycling of cellular components, plays a crucial role in maintaining cellular homeostasis [61]. In oncogenesis, the regulation of autophagy assumes a dual role in both suppressing and promoting tumor growth [61]. Consequently, depending on specific circumstances, the activation or inhibition of autophagy might offer therapeutic potential against advanced cancer [62].

Autophagy-associated genes (Atg) play a crucial role in the regulation of autophagy, a cellular process initiated by various stimuli such as nutrient scarcity. This process involves the formation of double-membrane vesicles known as autophagosomes, which encapsulate cytoplasmic components such as impaired or surplus proteins, organelles, lipids, and glycogen [63]. These components are often marked with ubiquitin and recognized by autophagy receptors such as p62 [63]. Cargo receptors facilitate cargo sequestration by interacting with both the cargo and the autophagosomal membrane constituent LC3-II [63]. Subsequently, the fusion of autophagosomes with lysosomes leads to the degradation of the cargo through the action of hydrolytic enzymes [63]. The flux involved in autophagy can be categorized as either unobstructed or impaired, with disruptions in autophagy flux gauged by monitoring the expression levels of the autophagy-specific substrate P62 [48]. Evidence indicates that PX treatment can induce the accumulation of the autophagy-specific substrates p62 and LC3-II in TNBC cells, suggesting that PX has the ability to induce autophagy and modulate its flux. Furthermore, co-administration of PX with an autophagy inhibitor resulted in increased suppression of TNBC cell lines, suggesting that PX may induce a protective autophagic response in TNBC.

Subsequent analysis of RNA-seq data following treatment with PX revealed significant changes in the PI3K-AKT-mTOR signaling pathway, which plays a crucial role in cell survival, proliferation, and metabolism and has been implicated in various cancers, including breast cancer [64,65,66,67]. Targeting the PI3K-AKT-mTOR axis is a promising therapeutic strategy for TNBC, with ongoing clinical trials supporting its potential efficacy [64]. Notably, mTOR represents a downstream effector of the PI3K-AKT pathway. Activation of AKT follows restructuring upon PI3K interaction with growth factor receptors, ultimately culminating in downstream effector pathway phosphorylation [67]. The mTOR signaling pathway is related to tumor cell survival. Our findings confirm the inhibitory effect of PX on the activation of AKT and mTOR. It is postulated that PX may exert its anticancer properties through modulation of this pathway.

The challenge of chemotherapy resistance poses a significant obstacle in cancer treatment, often resulting in patient relapse, often resulting in patient relapse. In recent years, combination drug therapies have emerged as a leading clinical approach, surpassing single-agent interventions in terms of efficacy and reduced toxicity. This shift towards combinatorial therapies has become a crucial aspect in the management of complex diseases thanks to network-based prediction of drug combinations. The PI3K-AKT-mTOR signaling pathway routinely harbors mutations across diverse cancers, leading to the development of treatment resistance [68,69,70,71]. Furthermore, combination treatment with PX and DOX has been demonstrated to yield inhibitory effects on TNBC at lower dosages, mitigating the long-term cardiac effects associated with high-dose DOX administration. Kosić et al. have highlighted the cardioprotective potential of DOX–PX synergy, which is characterized by reduced DOX-induced cardiotoxicity and improved survival rates, providing a grounded rationale for their conjoined deployment in mitigating the side effects of DOX [72].

In vivo effectiveness is a critical factor in assessing the anticancer properties of pharmaceuticals. In this study, we utilized a subcutaneous tumor model to evaluate the efficacy of PX combination therapy in reducing TNBC tumor growth. However, the addition of PX only slightly enhanced the inhibitory effects of chemotherapy. Despite subsequent analysis of the effects of combination therapy on TNBC lung metastasis, it is regrettable that this combined regimen did not successfully eradicate spontaneous lung metastasis derived from subcutaneous tumors (Figure 6F).

As a long-term clinically approved medication, the safety of PX is well-established, which is advantageous for its advancement into clinical trials for cancer treatment. As part of the safety evaluation for the combination regimen, we assessed the effects of drug combination on the body weight changes of tumor-bearing mice. Figure 6C demonstrates that neither the individual drug nor the drug combination resulted in significant harm to the mice. For mice weighing approximately 20 g, a daily dose of 20 mg/kg without changing the drug formulation corresponds to a human dose of 97.2 mg when calculated based on a body weight of 60 kg [73]. Previous clinical trials have shown that, contingent on the specific condition being treated, the total daily dosage of PX for adult psychiatric patients ranges from 20 to 50 mg [74,75,76]. It is worth noting that the dosage form of PX used in our current study differed from that used in clinical practice, potentially resulting in variations in pharmacokinetic parameters [77]. Therefore, the dose in our animal experiments may differ from the dose used in humans. Furthermore, in the event that the dosage of PX utilized in our research proves to be excessive for patients, it may be beneficial to refine drug delivery methods to specifically target the tumor site, thereby minimizing drug absorption in healthy tissues [78]. This approach could potentially result in decreased dosage requirements and improved therapeutic outcomes. Alternatively, conducting preclinical studies to identify the targets of PX in TNBC and exploring structural modifications of PX to enhance its anticancer activity while mitigating toxicity to normal tissues might be considered as well. Overall, it can be inferred that the relatively high dose of PX utilized in this study is not a barrier to its potential for repurposing in the treatment of TNBC. It is important to acknowledge that the tumor and circulating levels of PX were not quantified in our research, preventing us from ascertaining whether the systemic drug levels were adequate to elicit an antitumor response. In addition, the limited intratumoral drug concentrations may elucidate the lack of substantial efficacy of PX as a standalone treatment in vivo. Therefore, additional research will be undertaken to ascertain the intratumoral drug concentration, enhance drug delivery techniques for precise targeting of the tumor site [25], reduce the absorption rate in normal tissues, and increase the drug concentration in tumors, with the goal of ultimately augmenting the effectiveness of PX.

## 5. Conclusions

In conclusion, our study findings indicate that PX has potential as a therapeutic strategy through the activation of intrinsic apoptosis via mitochondrial pathways and the modulation of autophagy flux. Moreover, our research suggests that PX may inhibit TNBC by targeting the PI3K-AKT-mTOR signaling pathway. Our results could indicate a modest synergistic effect of PX when used in combination with chemotherapy in preclinical models, providing a basis for further exploration of personalized treatment strategies. Through a comprehensive analysis of both genotypic and phenotypic profiles of patients, the most suitable treatment regimen can be identified, ultimately improving the accuracy and efficacy of therapeutic interventions.

## Figures and Tables

**Figure 1 cancers-16-00885-f001:**
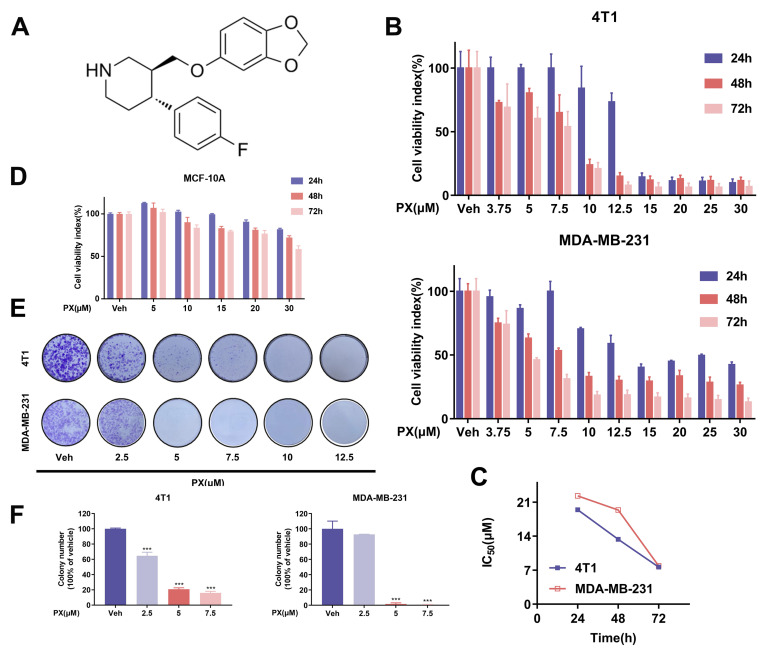
Inhibition of TNBC cell lines growth in vitro by PX. (**A**) The molecular structure of PX. (**B**) The antiproliferative efficacy of PX on TNBC cell lines 4T1 and MDA-MB-231 evaluated via MTT Assay. Cells were exposed to varied concentrations of PX (3.75–30 μM) for 24, 48, and 72 h (*n* = 3). (**C**) IC50 values of PX treatment determined for 4T1 and MDA-MB-231 cells at 24, 48, and 72 h. (**D**) PX exhibited inhibitory activity against the human non-tumorigenic breast cell lines MCF-10A at 24, 48, and 72 h (*n* = 3). (**E**) Assessment of colony formation capability of 4T1 and MDA-MB-231 cells after one week of PX treatment. (**F**) Quantification analysis of colony formation assay (*n* = 3). Data are presented as Mean ± SD. *** *p* < 0.001. Comparative analysis between groups utilized a two-tailed Student’s *t*-test or Mann–Whitney U test.

**Figure 2 cancers-16-00885-f002:**
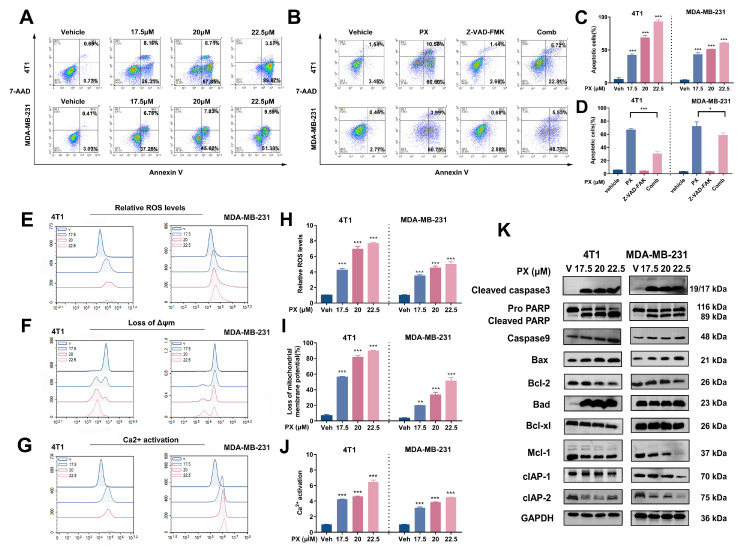
Induction of apoptosis in TNBC Cell lines by PX via activation of the mitochondria-mediated intrinsic pathway. (**A**) Apoptosis assessment in 4T1 cells and MDA-MB-231 cells following 24 h PX treatment. Cells were stained with Annexin V-PE and 7-AAD, then apoptotic events were quantified via flow cytometry. (**B**) Apoptosis evaluation in 4T1 cells and MDA-MB-231 cells upon treatment for 24 h with PX (20 μM) alone, Z-VAD-FMK (50 μM) alone, or their combined administration. Flow cytometry was utilized for detection of apoptosis. (**C**) Quantification analysis of apoptosis following PX treatment (*n* = 3). (**D**) Quantitative investigation of apoptosis induced by PX and Z-VAD-FMK (*n* = 3). (**E**) Relative ROS levels in 4T1 and MDA-MB-231 cells subjected to 24 h PX treatment. Cellular ROS levels were measured using DCFH-DA (10 μM) staining for 30 min and flow cytometry was used to assess ROS levels. (**F**) Level of Δψm disruption in 4T1 and MDA-MB-231 cell lines following 24 h PX treatment. Cells were stained with Rh123 (5 μg/mL) for 30 min and ΔΨm was analyzed via flow cytometry. (**G**) Intracellular Ca^2+^ concentration alterations in 4T1 and MDA-MB-231 cells post-24 h PX treatment. Fluo 3-AM (0.5 μM) staining for 30 min was employed for Ca^2+^ assessment by flow cytometry. (**H**) Quantification appraisal of relative ROS levels post-PX treatment (*n* = 3). (**I**) Quantification evaluation of ΔΨm disruption following PX treatment (*n* = 3). (**J**) Quantification analysis of intracellular Ca^2+^ concentration changes following PX treatment (*n* = 3). (**K**) Influence of varied PX concentrations on the expression of apoptosis-related proteins in 4T1 and MDA-MB-231 cell lines. The uncropped blots are shown in the Appendix A. Data are presented as mean ± S.D. *** *p* < 0.001, ** *p* < 0.01, * *p* < 0.05. Comparative analysis between groups utilized a two-tailed Student’s *t*-test or Mann–Whitney U test.

**Figure 3 cancers-16-00885-f003:**
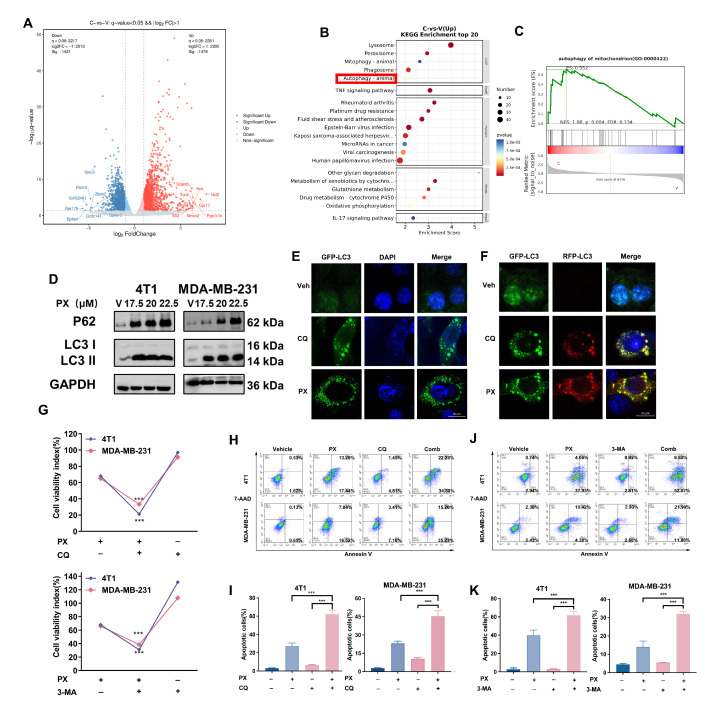
Induction of cytoprotective autophagy and autophagic flux inhibition by PX in TNBC cell lines. (**A**) Volcano plot depicting the DEGs identified from the RNA-seq data of 4T1 cells treated with or without PX (20 μM) over a 24 h interval. The PX Treatment group is represented by “C” and the Vehicle group is represented by “V”. (**B**) Results of KEGG pathway analysis, showing significant enrichment among the top 20 upregulated pathways. The PX Treatment group is represented by “C” and the Vehicle group is represented by “V”. (**C**) GSEA enrichment analysis results. (**D**) Expression of LC3 and P62 in 4T1 and MDA-MB-231 cells after PX treatment for 24 h. The uncropped blots are shown in the Appendix A. (**E**) Cells were transfected with the GFP-LC3 plasmid, revealing a representative image illustrating GFP-LC3 puncta formation subsequent to PX treatment. Scale bar: 20 μm (*n* = 3). (**F**) Presentation of representative images from 4T1 cells treated with PX and CQ subsequent to transfection with the tandem RFP-GFP-RFR tagged LC3B plasmid. Scale bar: 20 μm (*n* = 3). (**G**) Cell viability assay of 4T1 and MDA-MB-231 cells treated with PX (20 μM) for 24 h either with or without co-administration of CQ (20 μM) or 3-MA (2 mM) autophagy inhibitors (*n* = 3). (**H**) Assessment of apoptosis in 4T1 (17.5 μM) and MDA-MB-231 cells (17.5 μM) subsequent to treatment with PX and CQ (20 μM) individually or in combination for a duration of 24 h. (**I**) Quantitative analysis elucidating the extent of apoptosis induced by PX and CQ. (**J**) Assessment of apoptosis in 4T1 (17.5 μM) and MDA-MB-231 cells (15 μM) subsequent to treatment with PX and 3-MA (2 mM) individually or in combination for a duration of 24 h. (**K**) Quantitative analysis elucidating the extent of apoptosis induced by PX and 3-MA. Data are presented as mean ± S.D. *** *p* < 0.001. Statistical analyses employed the two-tailed Student’s *t*-test or Mann–Whitney U test for intergroup comparisons.

**Figure 4 cancers-16-00885-f004:**
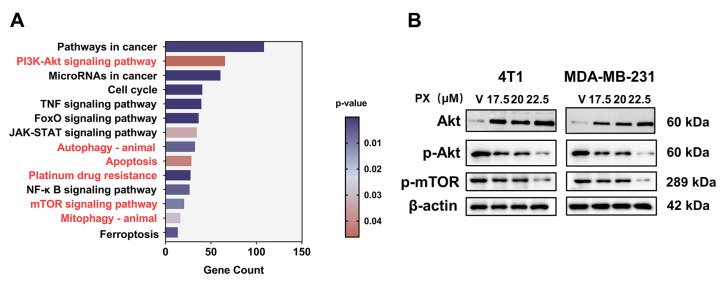
Inhibition of TNBC by PX via the PI3K-AKT-mTOR pathway. (**A**) KEGG analysis of DEGs. (**B**) Evaluation of expression levels of PI3K-AKT and mTOR pathway-associated proteins in 4T1 and MDA-MB-231 cells following exposure to varying concentrations of PX over a 24 h period. The uncropped blots are shown in the Appendix A.

**Figure 5 cancers-16-00885-f005:**
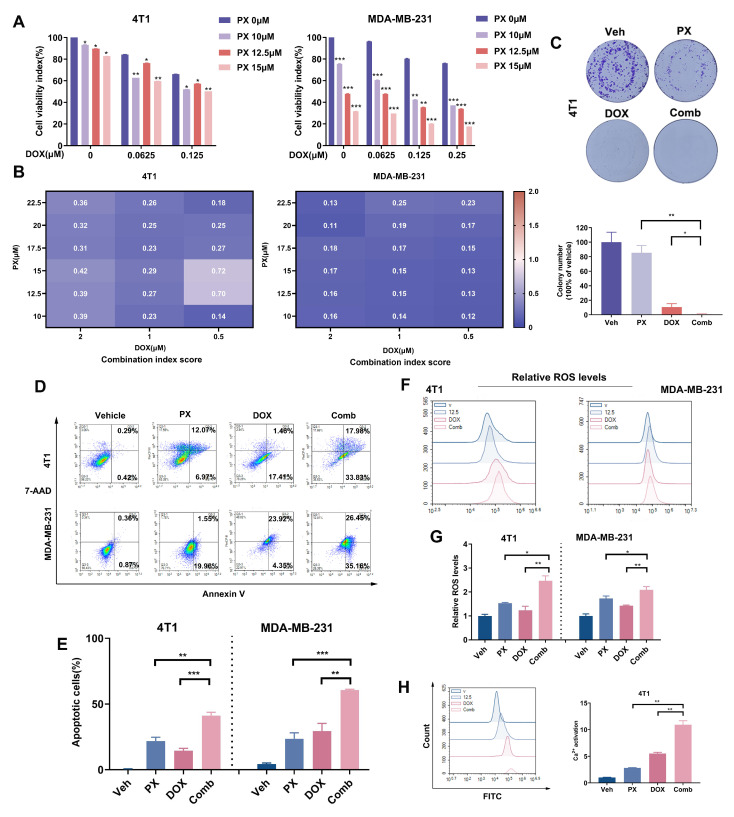
Synergistic effects of PX and DOX on viability and apoptosis in TNBC cell lines in vitro. (**A**) Dose-response of 4T1 and MDA-MB-231 cells to PX and DOX treatment alone and in combination was investigated at varying doses after 48 h of treatment. Cell viability was 100% for vehicle control (*n* = 3). (**B**) The CI values for the combination of PX and DOX were calculated using CompuSyn software (version 1.0 based on the inhibitory ratios of each drug combination dose. CI values less than 1 indicate synergy. (**C**) Quantification of colony formation in 4T1 cells following treatment with PX (2.5 μM), DOX (0.0625 μM), or combination thereof for a duration of one week. The results of the assay are depicted in the image below (*n* = 3). (**D**) Apoptosis in 4T1 and MDA-MB-231 cell lines treated with PX (12.5 μM) and DOX (0.125 μM) alone or in combination for 48 h. Apoptosis was detected by flow cytometry. (**E**) Quantitative study of apoptosis induced by PX and DOX (*n* = 3). (**F**) Evaluation of relative ROS levels in 4T1 and MDA-MB-231 cells following treatment with PX (12.5 μM), DOX (0.0625 μM), or their combination by 48 h. (**G**) Quantitative results of the relative ROS levels shown in (**F**) (*n* = 3). (**H**) Evaluation and quantification of Ca^2+^ concentration in 4T1 cells following treatment with PX (12.5 μM), DOX (0.0625 μM), or their combination for 48 h (*n* = 3). Data are expressed as mean ± S.D. *** *p* < 0.001, ** *p* < 0.01, * *p* < 0.05. Two-tailed Student’s *t*-test or Mann–Whitney U test used for comparison between groups.

**Figure 6 cancers-16-00885-f006:**
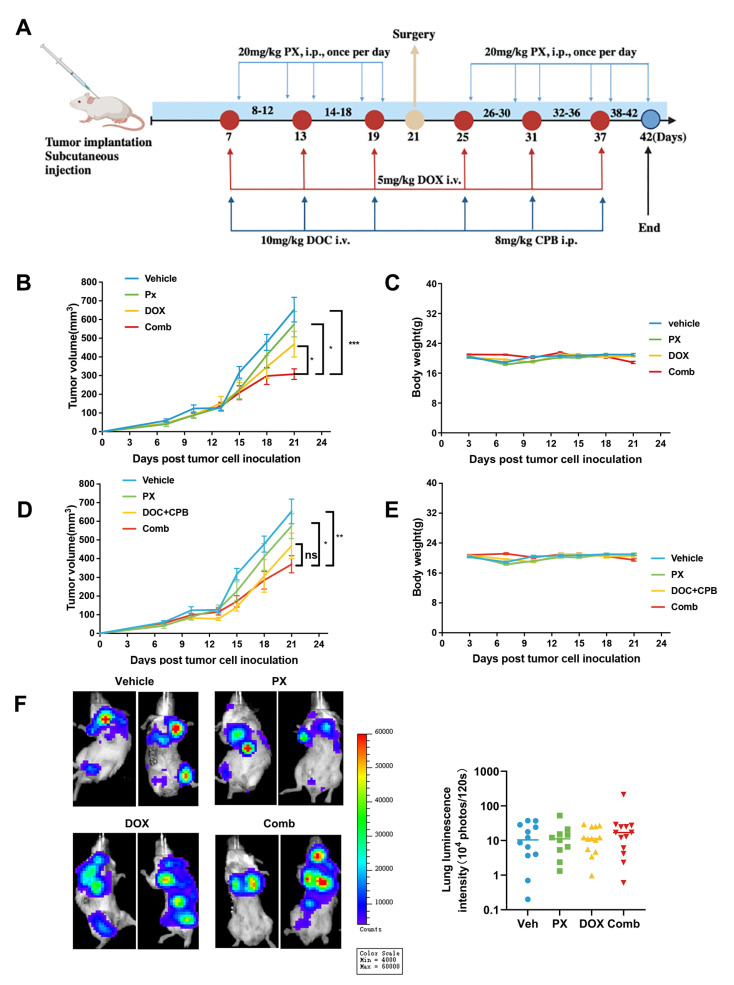
Synergistic impact of PX and chemotherapy on tumor growth in TNBC cells in vivo. (**A**) Illustration of the treatment regimen employed for subcutaneous inoculation of mice with 1 × 10^5^ 4T1-luc cells (*n* = 13). (**B**) Evaluation of the impact of PX, DOX, and their combined administration on tumor volume in mice. (**C**) Comprehensive monitoring of distinct treatments on the body weight trajectory of mice. (**D**) Evaluation of the impact of PX, CPB + DOC, and their combined administration on tumor volume in mice. (**E**) Comprehensive monitoring of distinct treatments on the body weight trajectory of mice. (**F**) Representative in vivo luminescence images capturing metastatic signals emanating from each experimental group post-tumor cell inoculation along with quantification of pulmonary luminescence intensity on day 42 following diverse treatment regimens (*n* = 9–12). Data are presented as mean ± S.D. Statistical significance is indicated as follows: *** *p* < 0.001, ** *p* < 0.01, * *p* < 0.05. Statistical analyses encompassed the two-tailed Student’s *t*-test or Mann–Whitney U test for group comparisons.

## Data Availability

Data will be made available on request.

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
