# Peer review of "Inhibition of TNBC Cell Growth by Paroxetine: Induction of Apoptosis and Blockage of Autophagy Flux"

_cancers, 2024, doi:10.3390/cancers16050885_

Round 1

Reviewer 1 Report

Comments and Suggestions for Authors

In the manuscript titled "Inhibition of TNBC Cell Growth via Remodeling of the Antidepressant Drug Paroxetine: Induction of Apoptosis and Blockage of Autophagy Flux through Mediation of the PI3K-AKT Pathway" by Qianrui Huang et al., the authors explore a critical and innovative approach within oncology by repurposing the antidepressant drug Paroxetine (PX) for treating triple-negative breast cancer (TNBC). The manuscript presents compelling findings, demonstrating the significant inhibition of 4T1 tumor growth in vitro and in vivo through a combination of PX and the chemotherapeutic drug doxorubicin (DOX). The elucidation of the molecular mechanisms, specifically how PX engages the Ca2+-dependent mitochondria-mediated intrinsic apoptosis pathway and induces cytoprotective autophagy via the PI3K-AKT pathway in TNBC, adds significant value to the field.

The concept of drug repurposing highlighted in this study is particularly commendable, offering a promising avenue to expedite the availability of new therapeutic options by leveraging existing drugs, potentially bypassing certain phases of drug development. The study's insights contribute meaningfully to the field and could bear significant clinical implications.

Nevertheless, upon meticulous examination, several aspects of the manuscript require further attention and clarification:

1.      Title Conciseness: The current title of the manuscript is somewhat verbose. A more concise title could more effectively capture the essence and key findings of the research.

2.      It's crucial to establish the clinical relevance of the PX dosage used in the in vivo experiments. Clarifying whether the dosage mirrors those used in clinical settings would add significant value to the study's applicability and validity.

3.      The manuscript does not sufficiently address why PX alone does not exhibit substantial efficacy in vivo. Providing a more detailed rationale for this observation and the subsequent synergistic effect observed in combination treatments, both in vitro and in vivo, would strengthen the study. Moreover, exploring the effectiveness of PX in combination with other chemotherapeutic agents for TNBC treatment could provide a broader perspective on its potential applications.

4.      To enhance the clinical relevance and applicability of the findings, it would be advantageous to include additional TNBC models in the in vivo experiments. This would provide a more comprehensive understanding of the potential impact of PX combined with chemotherapeutic drugs in TNBC treatment.

5.      The manuscript contains several typographical errors that need rectification for clarity and professionalism. For instance, "citecho2019paroxetine" in line 234 and “105 4T1-luc cells” in line 413 require correction.

Comments on the Quality of English Language

looks good. 

Author Response

In the manuscript titled "Inhibition of TNBC Cell Growth via Remodeling of the Antidepressant Drug Paroxetine: Induction of Apoptosis and Blockage of Autophagy Flux through Mediation of the PI3K-AKT Pathway" by Qianrui Huang et al., the authors explore a critical and innovative approach within oncology by repurposing the antidepressant drug Paroxetine (PX) for treating triple-negative breast cancer (TNBC). The manuscript presents compelling findings, demonstrating the significant inhibition of 4T1 tumor growth in vitro and in vivo through a combination of PX and the chemotherapeutic drug doxorubicin (DOX). The elucidation of the molecular mechanisms, specifically how PX engages the Ca2+-dependent mitochondria-mediated intrinsic apoptosis pathway and induces cytoprotective autophagy via the PI3K-AKT pathway in TNBC, adds significant value to the field.

The concept of drug repurposing highlighted in this study is particularly commendable, offering a promising avenue to expedite the availability of new therapeutic options by leveraging existing drugs, potentially bypassing certain phases of drug development. The study's insights contribute meaningfully to the field and could bear significant clinical implications.

Nevertheless, upon meticulous examination, several aspects of the manuscript require further attention and clarification:

1.Title Conciseness: The current title of the manuscript is somewhat verbose. A more concise title could more effectively capture the essence and key findings of the research.

Response:Thank you for your valuable suggestion. Based on your advice, we have revised the title of the article to better reflect its content: "Inhibition of TNBC Cell Growth by Paroxetine: Induction of Apoptosis and Blockage of Autophagy Flux."

2.It's crucial to establish the clinical relevance of the PX dosage used in the in vivo experiments. Clarifying whether the dosage mirrors those used in clinical settings would add significant value to the study's applicability and validity.

Response:Thank you for your valuable suggestion. According to the literature, a dosage of 20 mg/kg per day in mice weighing approximately 20 g, when converted to a human equivalent dose for a person weighing 60 kg using the formula mentioned in the literature, is 97.2 mg (doi: 10.4103/0976-0105.177703), which exceeds the typical clinical range of paroxetine use, which is 20-50 mg. However, the formulation of paroxetine used in the study differs from the dosage form used in clinical practice. Different formulations can lead to different pharmacokinetic parameters. Therefore, the dosage used in animal experiments may differ from the dosage used in human patients. We have included this discussion in our manuscript in lines 572-584. Thank you for your inquiry, which has contributed to the completeness of our manuscript.

3.The manuscript does not sufficiently address why PX alone does not exhibit substantial efficacy in vivo. Providing a more detailed rationale for this observation and the subsequent synergistic effect observed in combination treatments, both in vitro and in vivo, would strengthen the study.

Response:Thank you for your valuable suggestion. In our study, we did not measure the concentration of PX in mouse tumors or in circulation, so we cannot determine if the drug concentration in the body is sufficient to exert its anti-tumor effect. The low drug concentration in the tumor may also explain why using PX alone does not show significant efficacy in vivo. Therefore, we plan to measure the drug concentration in the tumor in further studies and improve drug delivery to enhance drug absorption and/or targeting at the tumor site. We have included this discussion in our manuscript in lines 590-597. Thank you for your inquiry, which has contributed to the completeness of our manuscript.

4.Moreover, exploring the effectiveness of PX in combination with other chemotherapeutic agents for TNBC treatment could provide a broader perspective on its potential applications.

Response:Thank you for your valuable suggestion. In the same animal experiment evaluating the efficacy of doxorubicin in combination with paroxetine, we conducted testing for the combination of docetaxel and carboplatin with paroxetine as well. Initially, we intended to incorporate the data of docetaxel and carboplatin in combination with paroxetine as supplementary data for another study., but due to this revision process, we have decided to present these data here. The corresponding data are now included in Fig. 6D-E. Once again, we appreciate the reviewer's suggestion, which has contributed to the completeness of our manuscript revision.

5.To enhance the clinical relevance and applicability of the findings, it would be advantageous to include additional TNBC models in the in vivo experiments. This would provide a more comprehensive understanding of the potential impact of PX combined with chemotherapeutic drugs in TNBC treatment.

Response:Thank you for your valuable suggestion regarding in vivo experiments. Incorporating additional TNBC models in in vivo experiments to validate the anticancer effects of paroxetine is indeed crucial. Such studies are essential for providing robust evidence for the translational potential of our findings. However, the process of ordering experimental animals, establishing other TNBC models, and obtaining corresponding results typically requires at least 40 days. Since the editor has given us only 10 days to complete this revision, we were unable to include animal experiments in this preliminary study. We fully acknowledge the necessity of further experiments in animal models to confirm the in vivo effects of paroxetine. Once again, we appreciate the reviewer's suggestion, which has contributed to the completeness of our manuscript revision.

6.The manuscript contains several typographical errors that need rectification for clarity and professionalism. For instance, "citecho2019paroxetine" in line 234 and “105 4T1-luc cells” in line 413 require correction.

Response:Thank you for your comment. We appreciate the reviewer's attention to detail and feedback on errors in our manuscript. We take these issues seriously and have conducted a thorough review of the entire paper, addressing and correcting all identified grammar and typographical errors. We are grateful to the reviewer for helping us improve the clarity and readability of our work. Your feedback is invaluable in enhancing the overall quality of the manuscript.

Reviewer 2 Report

Comments and Suggestions for Authors

In this study, Huang et. al investigated the inhibitory potential of paroxetine on triple-negative breast cancer (TNBC). The experimental design is sound and the experimental flow is well defined. The study assessed its inhibitory activity on 4T1 and MDA-MB-231 cells, its impact on apoptosis and autophagy, and explored potential target pathways. The intresting study provides rich data and provides a potential drug to treat TNBC. I have only minor comments about the present study. .

1.     In Figure 1, although Paroxetine shows potential anti-TNBC effects, the authors need to determine its IC50 concentration in non-tumor breast cell lines (e.g., MCF-10A) to study its impact on the proliferation of normal breast cells.

2.     The authors claim that paroxetine induces mitochondria-mediated intrinsic apoptosis. However, in Figure 2K, the authors did not show the changes in caspase 9 levels in paroxetine-treated TNBC cells, which is a marker of intrinsic apoptosis. It is recommended to include this result to strengthen the persuasiveness of the experimental findings.

3.     The authors suggest that paroxetine may induce changes in the PI3K/AKT pathway in TNBC. The PI3K/AKT/mTOR pathway is associated with autophagy-related pathways. The authors should investigate downstream pathways to provide sufficient evidence that paroxetine regulates this pathway, leading to autophagy.

4.     In Figure 4A, the authors demonstrate that the combination of paroxetine with CQ increases apoptosis in TNBC cells compared to single-drug treatment, indicating that paroxetine might induce protective autophagy. The occurrence of autophagy involves both the aggregation of autophagosomes and the fusion of autophagolysosomes. CQ is an inhibitor of autophagolysosome fusion. The authors should include experiments combining paroxetine with early-stage autophagy inhibitors to further support their claim.

Author Response

1.In Figure 1, although Paroxetine shows potential anti-TNBC effects, the authors need to determine its IC50 concentration in non-tumor breast cell lines (e.g., MCF-10A) to study its impact on the proliferation of normal breast cells.

Response:Thank you for your comment. To address this issue, we conducted MTT analysis using MCF-10A cells. In the updated Fig. 1D, our results demonstrate that, in comparison to TNBC cells, paroxetine exhibits lower toxicity towards MCF-10A cells. We have incorporated these findings into our study to enhance the comprehensiveness of the article.

2.The authors claim that paroxetine induces mitochondria-mediated intrinsic apoptosis. However, in Figure 2K, the authors did not show the changes in caspase 9 levels in paroxetine-treated TNBC cells, which is a marker of intrinsic apoptosis. It is recommended to include this result to strengthen the persuasiveness of the experimental findings.

Response:Thank you for your comment. In response to your suggestion, we have included the trend of caspase 9 in TNBC cells after paroxetine treatment in Fig. 2K. This addition serves to illustrate that paroxetine can activate the intrinsic pathway to induce apoptosis in TNBC cells. Your feedback has helped enhance the completeness of our manuscript, and we appreciate your thorough evaluation.

3.The authors suggest that paroxetine may induce changes in the PI3K/AKT pathway in TNBC. The PI3K/AKT/mTOR pathway is associated with autophagy-related pathways. The authors should investigate downstream pathways to provide sufficient evidence that paroxetine regulates this pathway, leading to autophagy.

Response:Thank you for your comment. In response to your suggestion, we have included the expression of p-mTOR proteins in Fig. 4B. This addition demonstrates that paroxetine can exert its inhibitory effect on TNBC by inhibiting the PI3K-AKT-mTOR pathway. Your feedback has contributed to enhancing the completeness of our manuscript, and we appreciate your thorough evaluation.

4. In Figure 4A, the authors demonstrate that the combination of paroxetine with CQ increases apoptosis in TNBC cells compared to single-drug treatment, indicating that paroxetine might induce protective autophagy. The occurrence of autophagy involves both the aggregation of autophagosomes and the fusion of autophagolysosomes. CQ is an inhibitor of autophagolysosome fusion. The authors should include experiments combining paroxetine with early-stage autophagy inhibitors to further support their claim.

Response:Thank you for your comment. In response to your suggestion, we have included the apoptosis flow cytometry experiment of TNBC cells treated with paroxetine in combination with the early autophagy inhibitor 3-MA in Fig. 3J-K. This addition demonstrates that paroxetine can synergistically inhibit TNBC with autophagy inhibitors and induce protective cell autophagy. Your feedback has contributed to enhancing the comprehensiveness of our manuscript, and we appreciate your thorough evaluation.

Reviewer 3 Report

Comments and Suggestions for Authors

Manuscript ID: 2848766

Type of manuscript: article

Title:  Inhibition of TNBC Cell Growth via Remodeling of the Antidepressant Drug Paroxetine: Induction of Apoptosis and Blockage of Autophagy Flux through Mediation of the PI3K-AKT Pathway

 Journal: Cancer

In this article, the authors have investigated the role of paroxetine (PX), a drug used for depression, with potential anticancer activity. In this study, they investigated the effects of PX on TNBC cells in vitro, both as a single agent and in combination with other drugs, using different methods. The findings indicate that PX trigger cytoprotective autophagy via the PI3K-AKT pathway in TNBC and therefore PX demonstrated significant inhibition of tumor growth in vivo.

The paper is well done and presented. All the criteria for a correct presentation were respected. The study is well described, and the results are conclusive and helpful.

I personally recommend the publication of this paper.

I have only a suggestion:

I suggest to describe better the reason in which used paroxetine in the context of this relevant research for example, in the introduction.

I suggest a title more simple and concise

Author Response

In this article, the authors have investigated the role of paroxetine (PX), a drug used for depression, with potential anticancer activity. In this study, they investigated the effects of PX on TNBC cells in vitro, both as a single agent and in combination with other drugs, using different methods. The findings indicate that PX trigger cytoprotective autophagy via the PI3K-AKT pathway in TNBC and therefore PX demonstrated significant inhibition of tumor growth in vivo. 

The paper is well done and presented. All the criteria for a correct presentation were respected. The study is well described, and the results are conclusive and helpful.

I personally recommend the publication of this paper.

I have only a suggestion:

I suggest to describe better the reason in which used paroxetine in the context of this relevant research for example, in the introduction.

I suggest a title more simple and concise.

Response:Thank you for your comment. In response to your suggestion, we have made a more suitable modification to the title of the article, specifically "Inhibition of TNBC Cell Growth by Paroxetine: Induction of Apoptosis and Blockage of Autophagy Flux," and incorporated the latest research advancements on paroxetine in the introduction (lines 71-85). Your feedback has contributed to enhancing the comprehensiveness of our manuscript, and we appreciate your thorough evaluation.

Reviewer 4 Report

Comments and Suggestions for Authors

The manuscript reports the use of a currently used SSRI Paroxetine as an anti-cancer agent in the setting of TNBC.  There are a number of major concerns about this study.  The first is that the concentrations of PX that have any cytotoxic effect are 10-100 above the effective clinical doses used as an antidepressant, and high doses have significant toxicities in patients.  This makes the use of this drug at these concentrations dangerous for patients.  The authors do a  good job of assess the mechanism of cytotoxicity of PX in their models, finding that they trigger a caspase dependent form of cell death and a paradoxical cytoprotective autophagy.  They show that inhibiting autophagy can enhance the cytotoxic effect of PX, although claim that this is through inhibition of PI3K-AKT.  However, they use an agent they claim inhibits PI3K to enhance the apoptosis induced by PX, so not sure how this PI3K inhibition can both enhance and inhibit cell death?

There are multiple other questions and comments detailed below.  

Please desist from the unnecessary use of non-standard abbreviations e.g. FCM for flow cytometry. 

The data in Fig 2 The data appears fine but it is very difficult to understand how the % apoptosis in 4T1 increases from 50% at 17.5 uM to 100% at 22.5uM.  The normal kinetics of inhibition usually require 10 fold increases in drug concentration to achieve IC90, here it is a 30% increase in drug concentration?  Similarly, the IC50 for PX in MDA231 is 22 uM, but in figure 2D 20uM PX induces >60% apoptosis in 24 h?  the very large changes in the various responses measured across very small changes in PX concentration suggests that the effect is not through usual enzyme inhibition kinetics, although as PX appears to interact and inhibit a wide variety of enzymes and kinases this might be an explanation for this non-standard dose response.  

Fig 2; The effects on pro- and anti-apoptotic BCL2-related and BH3 only proteins and IAPs is related to the increased apoptosis, but there appears to be little changes in these yet especially in 4T1 very large changes in apoptosis. As high concentrations of z-VAD  only modestly reduced cell death, is a large part of the cell death caspase independent?  

No experimental details provided about the RNAseq experiments.  How many replications, what methods used?  

There is no point marking the enrichment score for autophagy on 3B.  And why autophagy when lysosome was by far the highest enrichment score and number of genes? 

 Fig 4D; The data shows that PX treatment reduces pAKT level, but  the directionality of the decrease changes between cell lines, 4T1 shows continuous decrease with increasing drug dose whereas MDA-231 is the opposite.  Is this reproducible and what does this mean?  The linkage to autophagy is fine although it must be experimentally validated in this system, especially as the effects of PX are so broad in terms of its potential cellular targets.  Additionally, you have used 3MA to inhibit PI3K -dependent autophagy to improve apoptosis, but then claim that PX inhibition of PI3K-AKT is a beneficial outcome. How does this work both ways, inhibiting and enhancing cell death similtaneously?  

Fig 5; the data for the combination at least in the change sin viability and apoptosis are little more than additive.  The combination index data are for much high concentrations of of both drugs than shown in Fig 5A, so these are already likely to be highly toxic as single agents, potentially making the calculations of combination index inaccurate or relatively meaningless. 

There is confusion about where the 4T1 cells were initially implanted, in the left abdominal cavity or subcutaneously as described in the Methods?  

Comments on the Quality of English Language

The language used is excessively florid rather than concise explanation of the points raised.  It makes reading the manuscript very difficult.  I recommend external editorial assistance to improve the language.  

Author Response

1. The manuscript reports the use of a currently used SSRI Paroxetine as an anti-cancer agent in the setting of TNBC.  There are a number of major concerns about this study.  The first is that the concentrations of PX that have any cytotoxic effect are 10-100 above the effective clinical doses used as an antidepressant, and high doses have significant toxicities in patients.  This makes the use of this drug at these concentrations dangerous for patients.

Response:Thank you for your valuable suggestion. According to the literature, a dosage of 20 mg/kg per day in mice weighing approximately 20 g is equivalent to 97.2 mg for a person weighing 60 kg, using the formula mentioned in the literature (doi: 10.4103/0976-0105.177703). This dosage only exceeds the clinically used dose by 1-3 times, which is 20-50 mg. However, the formulation of paroxetine used in the study differs from the dosage form used in clinical practice, leading to potential differences in pharmacokinetic parameters. Therefore, the dosage form used in animal experiments may not directly translate to human patients.

We conducted a safety assessment for the mouse combination regimen, evaluating the impact of drug combinations on the body weight changes of tumor-bearing mice. Our analysis of the body weight data indicated that neither the individual drug nor the drug combination caused significant harm to the mice.

Additionally, if the dosage of PX in our study is indeed too high for patients, we can optimize the drug delivery route, similar to the optimization of doxorubicin dosage form (doi: 10.2165/00003088-200342050-00002; doi: 10.7314/apjcp.2014.15.1.489), by directly delivering the drug to the tumor site to reduce systemic drug exposure. Alternatively, modifying the compound based on the target of PX in TNBC may enhance its anticancer activity and reduce toxicity to normal tissues. These are areas we plan to research further. Therefore, our conclusion is that the relatively high dose of paroxetine in this study does not preclude its potential repurposing in TNBC. We have included this discussion in our manuscript in lines 572-590. Thank you for your inquiry, which has contributed to the completeness of our manuscript.

2. The authors do a good job of assess the mechanism of cytotoxicity of PX in their models, finding that they trigger a caspase dependent form of cell death and a paradoxical cytoprotective autophagy.  They show that inhibiting autophagy can enhance the cytotoxic effect of PX, although claim that this is through inhibition of PI3K-AKT.  However, they use an agent they claim inhibits PI3K to enhance the apoptosis induced by PX, so not sure how this PI3K inhibition can both enhance and inhibit cell death?

Response:Thank you for your insightful suggestions, which demonstrate a high level of professionalism and significantly contribute to enhancing the logical coherence of this manuscript. Our current hypothesis posits that PX may exert cytotoxic effects by suppressing the PI3K-AKT pathway. The administration of PX initiates cytoprotective autophagy, which may be linked to its influence on other target proteins. Both 3-MA and CQ function as early and late autophagy inhibitors, respectively. Our findings suggest that they can enhance the activity of PX, indicating that autophagy inhibition augments their anticancer efficacy. While 3-MA acts as a PI3K inhibitor and PX demonstrates inhibition of PI3K-AKT, PX may induce cytoprotective autophagy through alternative pathways. Therefore, the addition of any autophagy inhibitor, whether it be a PI3K inhibitor like 3-MA or CQ, to PX could enhance its effectiveness. Similar effects have been observed with various compounds, wherein they induce protective autophagy alongside cytotoxicity by inhibiting the PI3K-AKT-mTOR pathway (e.g., "Quercetin induces protective autophagy in gastric cancer cells: Involvement of Akt-mTOR- and hypoxia-induced factor 1α-mediated signaling"; "Ivermectin induces cytostatic autophagy by blocking the PAK1/Akt axis in breast cancer"). We appreciate your feedback, and corresponding descriptions in the manuscript have been revised accordingly.

3. Please desist from the unnecessary use of non-standard abbreviations e.g. FCM for flow cytometry. 

Response:Thank you for your response. We have reviewed and corrected unnecessary abbreviations, and we appreciate your advice in improving the readability of our article.

4. The data in Fig 2 The data appears fine but it is very difficult to understand how the % apoptosis in 4T1 increases from 50% at 17.5 uM to 100% at 22.5uM.  The normal kinetics of inhibition usually require 10 fold increases in drug concentration to achieve IC90, here it is a 30% increase in drug concentration? 

Response:Thank you for your comment. The pharmacological effects of drugs typically correlate with their doses within a specific range. However, we have observed that the inhibitory effects on tumor cells may not follow a linear relationship with the concentration of different compounds. For instance, taxane compounds exhibit nonlinear pharmacokinetics over a short period (doi: 10.1007/s40262-017-0563-z). Moreover, the activity of some compounds undergoes significant changes after surpassing a certain threshold. Despite repeating the experiments, we consistently arrived at similar conclusions. The specific target of PX in inhibiting TNBC remains unidentified, and different targets may yield distinct effects, thus classical effects may not manifest.

5. Similarly, the IC50 for PX in MDA231 is 22 uM, but in figure 2D 20uM PX induces >60% apoptosis in 24 h? the very large changes in the various responses measured across very small changes in PX concentration suggests that the effect is not through usual enzyme inhibition kinetics, although as PX appears to interact and inhibit a wide variety of enzymes and kinases this might be an explanation for this non-standard dose response.

Response:This may be due to the different methods used in various experiments. The MTT assay principle involves succinate dehydrogenase in active cell mitochondria reducing exogenous MTT to insoluble blue-purple formazan crystals, which deposit in cells and are dissolved by DMSO. Dead cells lack this function. Absorbance is measured at 570nm wavelength using an enzyme immunoassay, and the absorbance value can be used to determine the number of live cells, with higher OD values indicating more live cells. In apoptosis detection, we use Annexin V and 7-AAD as apoptosis dyes. Annexin V selectively binds to phosphatidylserine (PS). During early apoptosis, PS is externalized to the cell surface. FITC-labeled Annexin V can bind to externalized PS, thus detecting early apoptosis. 7-AAD is a DNA-binding dye that stains the nuclei of necrotic cells or cells in late apoptosis with compromised cell membrane integrity, measuring fully dead cells. The flow cytometry plot in Figure 2D actually includes both early and late apoptotic cells, with approximately 45.62% early apoptotic cells at 20uM. Early apoptotic cells have only experienced cell membrane damage, and they are likely partially adherent and detected by MTT. Therefore, it is possible that the results you described occurred at concentrations lower than the MTT-determined IC50.

6. Fig 2; The effects on pro- and anti-apoptotic BCL2-related and BH3 only proteins and IAPs is related to the increased apoptosis, but there appears to be little changes in these yet especially in 4T1 very large changes in apoptosis. As high concentrations of z-VAD only modestly reduced cell death, is a large part of the cell death caspase independent?  

Response:Thank you for your comment. We completely agree with your viewpoint. There appear to be minimal changes in these aspects, particularly in 4T1, where there are substantial changes in apoptosis. Additionally, there may be caspase-independent apoptotic induction mediated by PX. We have already incorporated these modifications into the manuscript, as evidenced by the changes in lines 267-268, which indicate that this effect is only partially mediated through caspase-dependent apoptotic induction.

7. No experimental details provided about the RNAseq experiments.  How many replications, what methods used?  

Response:Thank you for your valuable suggestions. We have elaborated on the procedures for RNAseq experiments in section 2.11, as outlined in lines 176-185. Your input has been greatly appreciated and will enhance the publication of our article.

8. There is no point marking the enrichment score for autophagy on 3B.  And why autophagy when lysosome was by far the highest enrichment score and number of genes? 

Response:Thank you for your feedback. The enrichment score for autophagy in Figure 3B is approximately 2, indicating the relevance and significance of our focus on autophagy. Furthermore, the process of complete autophagy entails the aggregation of autophagosomes and their subsequent fusion with lysosomes, as elaborated in the text. The prominence of lysosomes, as indicated by their highest enrichment score and number of genes, further supports the induction of autophagy.

9. Fig 4D; The data shows that PX treatment reduces pAKT level, but the directionality of the decrease changes between cell lines, 4T1 shows continuous decrease with increasing drug dose whereas MDA-231 is the opposite.  Is this reproducible and what does this mean? 

Response:Thank you for your valuable suggestions. We have reviewed the replicated experiments and confirmed that the levels of pAKT in PX-treated MDA-MB-231 cells are consistent with those in the 4T1 cell lines, demonstrating a decreasing trend. We have accordingly corrected Figure 4B in the manuscript. Your assistance in rectifying the errors in our manuscript is greatly appreciated, as it is crucial for its publication.

10. The linkage to autophagy is fine although it must be experimentally validated in this system, especially as the effects of PX are so broad in terms of its potential cellular targets.  Additionally, you have used 3MA to inhibit PI3K -dependent autophagy to improve apoptosis, but then claim that PX inhibition of PI3K-AKT is a beneficial outcome. How does this work both ways, inhibiting and enhancing cell death similtaneously?

Response:Thank you for your insightful suggestions; they were of high professionalism and significantly contributed to enhancing the logical coherence of this manuscript. Presently, we hypothesize that PX may elicit cytotoxic effects by suppressing the PI3K-AKT pathway. Administration of PX initiates cytoprotective autophagy, potentially linked to its influence on other target proteins. Both 3-MA and CQ act as early and late autophagy inhibitors, respectively. Our findings suggest that they can amplify PX's activity, implying that autophagy inhibition enhances their anticancer efficacy. While 3-MA acts as a PI3K inhibitor and PX demonstrates inhibition of PI3K-AKT, PX may induce cytoprotective autophagy through alternative pathways. Thus, the addition of any autophagy inhibitor, whether it be a PI3K inhibitor like 3-MA or CQ, to PX could enhance its effectiveness. Similar effects have been observed with various compounds, wherein they induce protective autophagy alongside cytotoxicity by inhibiting the PI3K-AKT-mTOR pathway(Quercetin induces protective autophagy in gastriccancer cells: Involvement of Akt-mTOR- and hypoxia-induced factor 1α-mediated signaling,Ivermectin Induces Cytostatic Autophagy by Blocking the PAK1/Akt Axis in Breast Cancer). We have already made corresponding changes in the manuscript, such as in lines 372-382 and 541-544. We value your feedback; corresponding descriptions in the manuscript have been revised accordingly.

11. Fig 5; the data for the combination at least in the change sin viability and apoptosis are little more than additive.  The combination index data are for much high concentrations of of both drugs than shown in Fig 5A, so these are already likely to be highly toxic as single agents, potentially making the calculations of combination index inaccurate or relatively meaningless. 

Response:Thank you for your comment; your point is valid. However, we observed significant synergistic effects even at low concentrations of the combination in MDA-MB-231 cells (Fig. 5A), which is meaningful. The revision time provided by the editor is short, but we will conduct further experiments in additional TNBC cell lines and at various concentrations, particularly in the lower concentration range, to assess the combined effects of PX and multiple chemotherapy drugs.

12. There is confusion about where the 4T1 cells were initially implanted, in the left abdominal cavity or subcutaneously as described in the Methods?  

Response:We appreciate the reviewer's attention to detail and feedback on the inaccuracies in our manuscript. The establishment of the 4T1 subcutaneous tumor model involved subcutaneous injection of 4T1-luc cells on the left flank of the mice, which was incorrectly described in the manuscript (lines 201 and 440). We have made the necessary corrections. Thank you very much for helping us improve the clarity and readability of our work. Your feedback is highly valuable in enhancing the overall quality of the manuscript.

Reviewer 5 Report

Comments and Suggestions for Authors

Manuscript ID : Cancers-2848766

The Manuscript " Inhibition of TNBC Cell Growth via Remodeling of the Antidepressant Drug Paroxetine: Induction of Apoptosis and Blockage of Autophagy Flux through Mediation of the PI3K-AKT Pathway" Yiwen Zhang et al

In this Article the authors carefully drafted this manuscript on Drug repurposing to identify new applications on existing medications. such as Paroxetine, a commonly prescribed antidepressant, has exhibited potential anticancer effects in preclinical studies. they also shows Paroxetine has exhibited synergistic attributes when employed in conjunction with DOX in both in vitro and in vivo assessments.

Authors have well documented  with all supporting information and provide sufficient citation in the manuscript.

This manuscript holds valuable information for scientific readers and is suitable for publication.

Author Response

The Manuscript " Inhibition of TNBC Cell Growth via Remodeling of the Antidepressant Drug Paroxetine: Induction of Apoptosis and Blockage of Autophagy Flux through Mediation of the PI3K-AKT Pathway" Yiwen Zhang et al

In this Article the authors carefully drafted this manuscript on Drug repurposing to identify new applications on existing medications. such as Paroxetine, a commonly prescribed antidepressant, has exhibited potential anticancer effects in preclinical studies. they also shows Paroxetine has exhibited synergistic attributes when employed in conjunction with DOX in both in vitro and in vivo assessments.

Authors have well documented with all supporting information and provide sufficient citation in the manuscript.

This manuscript holds valuable information for scientific readers and is suitable for publication.

Response:Thank you for your comment. We will continue to revise the manuscript based on the comments and suggestions from other reviewers, aiming to meet their requirements. Your feedback has contributed to enhancing the completeness of our manuscript, and we appreciate your thorough evaluation.

Round 2

Reviewer 1 Report

Comments and Suggestions for Authors

The authors have provided additional data and made improvements that bolster their conclusions, partially addressing my review comments within the constrained timeframe.

Comments on the Quality of English Language

looks good. 

Author Response

The authors have provided additional data and made improvements that bolster their conclusions, partially addressing my review comments within the constrained timeframe.

Response:Thank you for your constructive comments and suggestions on our manuscript. We are pleased to learn that you have acknowledged the additional data we provided and the progress we have made in strengthening our conclusions. We will continue to revise the manuscript based on the comments and suggestions from other reviewers, aiming to meet their requirements. Your feedback has contributed significantly to enhancing the completeness of our manuscript, and we appreciate your thorough evaluation.

Reviewer 4 Report

Comments and Suggestions for Authors

The authors have addressed many of my comments and questions from the initial review and made changes that improve their manuscript.  However, one issue remains and that is reported synergy of their PX treatment with Dox.  I agree that the the combination is an improvement over the individual treatments, but at no point does this appear to be the robust synergy as claimed.   I strongly suggest the author reconsider this claim as there is no strong data to support it.  

Comments on the Quality of English Language

It is fine

Author Response

The authors have addressed many of my comments and questions from the initial review and made changes that improve their manuscript. However, one issue remains and that is reported synergy of their PX treatment with Dox. I agree that the the combination is an improvement over the individual treatments, but at no point does this appear to be the robust synergy as claimed. I strongly suggest the author reconsider this claim as there is no strong data to support it.

Response:Thank you for your insightful suggestions; they were highly professional and significantly contributed to enhancing the logical coherence of this manuscript. We fully acknowledge the issues raised by the reviewer. While the data in the manuscript do demonstrate the synergistic capability of PX treatment with DOX, this does not necessarily indicate a strong synergistic effect of the combined PX and DOX therapy. We have already made modifications in the manuscript. In the revised document, we have replaced descriptions such as "exhibiting strong synergy" with more objective statements, such as "the combination of PX and DOX demonstrates some synergistic effects." (lines 466, 500, 530-538, 701-702, 742-743) to address this. Your feedback has contributed significantly to enhancing the completeness of our manuscript, and we appreciate your thorough evaluation.
